# Price negotiation and pricing of anticancer drugs in China: An observational study

Jing Zhou[1,2], Tianjiao Lan[1,2], Hao Lu[3], Jay Pan[1,4]*

1 HEOA Group, West China School of Public Health and West China Fourth Hospital, Sichuan University, Chengdu, China, 2 Institute for Healthy Cities and West China Research Center for Rural Health Development, Sichuan University, Chengdu, China, 3 School of Public Health, Imperial College London, London, United Kingdom, 4 School of Public Administration, Sichuan University, Chengdu, China

* panjie.jay@scu.edu.cn

## Abstract

### Background

While China has implemented reimbursement-linked drug price negotiation annually since 2017, emphasizing value-based pricing to achieve a value-based strategic purchase of medical insurance, whether drug prices became better aligned with clinical value after price negotiation has not been sufficiently established. This study aimed to assess the changes in prices and their relationship with the clinical value of anticancer drugs after the implementation of price negotiations in China.

### Methods and findings

In this observational study, anticancer drug indications that were negotiated successfully between 2017 and 2022 were identified through National Reimbursement Drug Lists (NRDL) of China. We excluded extensions of indications for drugs already listed in the NRDL, indications for pediatric use, and indications lacking corresponding clinical trials. We identified pivotal clinical trials for included indications by consulting review reports or drug labels issued by the Center for Drug Evaluation, National Medical Products Administration. We calculated treatment costs as outcome measures based on publicly available prices and collected data on clinical value including safety, survival, quality of life, and overall response rate (ORR) from publications of pivotal clinical trials. The associations between drug costs and clinical value, both before and after negotiation, were analyzed using regression analyses. We also examined whether price negotiation has led to a reduction in the variation of treatment costs for a given value.

We included 103 anticancer drug indications, primarily for the treatment of blood cancer, lung cancer, and breast cancer, with 76 supported by randomized controlled trials and 27 supported by single-arm clinical trials. The median treatment costs over the entire sample have been reduced from US$34,460.72 (interquartile range (IQR): 19,990.49 to 55,441.66) to US$13,688.79 (IQR: 7,746.97 to 21,750.97) after price negotiation (P < 0.001). Before price negotiation, each additional month of survival gained was associated with an increase in treatment costs of 3.4% (95% confidence interval (CI) [2.1, 4.8], P < 0.001) for indications

**Data Availability Statement:** All relevant data are within the manuscript and its Supporting Information files.

**Funding:** This work was supported by the National Natural Science Foundation of China (72374149 and 72074163 to JP), Sichuan Science and Technology Program (2022YFS0052 and

2021YFQ0060 to JP), and China Center for South Asian Studies, Sichuan University (to JP). The funders had no role in study design, data collection and analysis, decision to publish, or preparation of the manuscript.

**Competing interests:** The authors have declared that no competing interests exist.

**Abbreviations:** CDE, Center for Drug Evaluation; CI, confidence interval; EPR, external price referencing; HTA, health technology assessment; IQR, interquartile range; MEA, managed entry agreement; MOHRSS, Ministry of Human Resources and Social Security; NHSA, National Healthcare Security Administration; NMPA, National Medical Products Administration; NRDL, National Reimbursement Drug Lists; ORR, overall response rate; OS, overall survival; PFS, progression-free survival; QoL, quality of life.

supported by randomized controlled trials, and a 10% increase in ORR was associated with a 6.0% (95% CI [1.6, 10.3], $P = 0.009$) increase in treatment costs for indications supported by single-arm clinical trials. After price negotiation, the associations between costs and clinical value may not have changed significantly, but the variation of drug costs for a given value was reduced. Study limitations include the lack of transparency in official data, missing data on clinical value, and a limited sample size.

## Conclusions

In this study, we found that the implementation of price negotiation in China has led to drug pricing better aligned with clinical value for anticancer drugs even after substantial price reductions. The achievements made in China could shed light on the price regulation in other countries, particularly those with limited resources and increasing drug expenditures.

## Author summary

### Why was this study done?

➢ China has implemented reimbursement-linked drug price negotiation annually since 2017, with an emphasis on value-based pricing to realize value-based strategic purchase of medical insurance.

➢ However, whether drug prices became better aligned with clinical value after the implementation of price negotiation has not been sufficiently established.

### What did the researchers do and find?

➢ For 103 included anticancer drug indications successfully negotiated between 2017 and 2022, identified through the National Reimbursement Drug Lists (NRDL) of China, we calculated treatment costs based on publicly available prices and collected data on clinical value including safety, survival, quality of life, and overall response rate (ORR) from publications of pivotal clinical trials.

➢ We employed regression analyses to explore the relationship between drug costs and clinical value both before and after price negotiation. Additionally, we investigated whether price negotiation resulted in a reduction in the variation of treatment costs for a given value.

➢ The median treatment costs for anticancer drug indications decreased from $34,460.72 (IQR: 19,990.49 to 55,441.66) before negotiation to $13,688.79 (IQR: 7,746.97 to 21,750.97) after negotiation ($P < 0.001$).

➢ Greater clinical value of anticancer drug indications was positively associated with higher treatment costs before price negotiation. After negotiation, this association remained largely unchanged, but the variation of treatment costs for a given value was reduced.

**What do these findings mean?**

➢ These data suggest that the implementation of price negotiation has not only resulted in substantial price reductions but has also led to better alignment of drug pricing with the clinical value of anticancer drugs.

➢ For countries grappling with increasing drug expenditures and limited resources, China could serve as an example of how price negotiation can be structured to better align prices with clinical value, in addition to price reductions.

➢ Study limitations include the lack of transparency in official data, missing data on clinical value, and a limited sample size.

## Introduction

The escalating global cancer burden and unmet clinical needs have contributed to a significant surge in research and development of anticancer treatments over the past decades [1,2]. Driven by continued innovation, worldwide spending on anticancer drugs reached $185 billion in 2021 and is projected to exceed $300 billion by 2026 [3]. China, being the largest developing country, has optimized its regulatory policies for innovative drugs, leading to the launch of numerous anticancer drugs in recent years [4–6]. Consequently, expenditure on anticancer drugs in China increased from $4.8 billion in 2016 to $13 billion in 2021, marking an increase of $8.2 billion [3].

The financial toxicity of high-priced anticancer drugs is now a well-recognized issue receiving widespread attention even across developed countries [7]. To curtail drug prices, ensure affordable patient access, and safeguard the sustainability of healthcare systems, health authorities use a mix of policy instruments in pricing and reimbursement [8]. In addition to external price referencing (EPR), managed entry agreements (MEA), and health technology assessment (HTA), value-based pricing and strategic procurement are being explored internationally including in China [8–11]. In China, such reforms were formally introduced in the form of reimbursement-linked drug price negotiation in 2017 (S1 Text). Since then, health authorities have been negotiating prices for innovative drugs directly with pharmaceutical companies annually, informed by HTA and accompanied by the EPR and MEA, trying to realize value-based strategic purchase of medical insurance [5,12]. If an agreement on the reimbursement price is reached between the 2 sides during the negotiation stage, the candidate drug becomes eligible for inclusion in the National Reimbursement Drug List (NRDL).

The price negotiation in China is still in its early stages and constantly evolving. From 2018 onward, the newly established National Healthcare Security Administration (NHSA) took over the responsibility of the price negotiation from the Ministry of Human Resources and Social Security (MOHRSS). For candidate drugs qualified for negotiation, the NHSA's target prices are determined by 2 parallel groups consisting of public medical insurance executives and pharmacoeconomics experts, respectively [5]. The former estimates the prices based on the pre-negotiation prices and the sustainability of the insurance fund. Meanwhile, the latter focuses on the comparative effectiveness and safety of the candidates compared to existing treatments, and assesses the pharmacoeconomic reports and budget impact analysis submitted by pharmaceutical companies using domestic and international prices as references, in an attempt to align prices more closely with clinical benefits [5,13]. In the context of limited

resources, rationalizing the relationship between resource inputs and outcomes can enhance the efficiency of countries' healthcare systems. Therefore, optimizing the alignment of prices and clinical value has great potential to maximize health outcomes under resource constraints from a societal perspective and incentivize the development of clinically meaningful drugs [14].

The clinical evidence considered by NHSA experts includes systematic reviews, meta-analyses, randomized controlled trials, and single-arm clinical trials. We searched for systematic reviews or meta-analyses providing high-level clinical evidence for anticancer drugs at the time of negotiation. We found that a considerable proportion of anticancer drugs lacked supportive systematic reviews or meta-analyses during the price negotiation period. Moreover, for those anticancer drugs with available systematic reviews or meta-analyses, the heterogeneity within study populations contradicted the intention of both the regulatory agency and the NHSA to specify and restrict the targeted population. Therefore, pivotal clinical trials supporting the approval of anticancer drugs are likely to be the most crucial evidence basis for decision-making during the price negotiation process.

Previous studies on price negotiation in China have primarily evaluated its impact on the expenditure, volume, availability, and accessibility of anticancer drugs [15–18]. While the NHSA lays stress on the value-based pricing of innovative drugs during the price negotiation process, whether the implementation of price negotiation has resulted in a better alignment of prices with the clinical value of anticancer drugs in China has not been sufficiently established. Furthermore, there is a lack of international studies examining the impact of pricing-related policy on the relationship between prices and clinical value [19]. To fill these gaps, we aimed to assess the post-implementation changes in prices and their relationship with the clinical value of anticancer drugs in China based on clinical evidence derived from pivotal clinical trials. We hypothesize that price negotiation has led to drug pricing being better aligned with clinical value in China.

## Methods

### Sample selection

This study identified anticancer drugs along with their indications that underwent price negotiation successfully between 2017 and 2022 using publicly available NRDL [20–25]. The selection of 2017 as the initial year corresponds to the initiation of price negotiation. We excluded extensions of indications for drugs already listed in the NRDL. We also excluded one indication for pediatric use to mitigate the difference among indications and ensure the consistency of our sample. Indications that were withdrawn from the NRDL but had previously been listed through price negotiation were included since the negotiated prices were reached at the time of price negotiation and were publicly available. Ethical approval was not required for this study as human subjects were not involved.

### Data sources and extraction

We extracted and reviewed review reports and drug labels from the website of the Center for Drug Evaluation, National Medical Products Administration (CDE, NMPA), and supplemented the information with a search of the Chinese pharmaceutical database, Drugdataexpy, when necessary [26,27]. Indications without matching clinical trials in their labels were excluded. We identified the pivotal clinical trial for each therapeutic indication in the section of "pivotal studies" in the review report. In cases where the review report was not available, we referred to the drug label. Eligible trials included single-arm clinical trials and randomized controlled trials that had an experimental group using the drug in our sample and a control group using its comparator, regardless of the number of arms. When trials featured multiple

experimental arms, we selected the arm that best aligned with the approved indication in the drug label. In cases where multiple trials were relevant, we sequentially chose the trial that best matched the indication, targeted the Chinese or Asian population, or had the best clinical outcome [28].

To assess the clinical value of included therapeutic indications, we extracted information on safety, overall survival (OS), progression-free survival (PFS), and quality of life (QoL) of randomized controlled trials, as well as overall response rates (ORRs) of single-arm clinical trials from peer-reviewed publications [29–31]. The extracted data were limited to those available at the time of price negotiation. For indications supported by randomized controlled trials, we collected safety measures, including the incidence of all adverse events, grade 3/4 adverse events, and serious adverse events [29]. After a comprehensive evaluation of these measures, we categorized the comparative safety of the study drug in relation to the comparator into 2 categories: improvement or no difference, and reduction. We extracted the median OS in the experimental and control arms, or PFS when trials did not report OS. Then, we calculated the absolute difference as added survival benefits between the 2 arms for each therapeutic indication [19,32]. We excluded indications for which pivotal clinical trials did not have OS or PFS as clinical endpoints or did not report OS or PFS in median times. The QoL was categorized into 3 categories: improvement, no difference, and reduction or unavailability, after reviewing the relevant contents in the publications for the pivotal trials [29]. For indications supported by single-arm clinical trials, we only extracted the ORR as the clinical value indicator because no comparator was involved. Two researchers with backgrounds in pharmacy independently analyzed the results, and consensus was reached through discussion in cases of disagreement.

We retrieved negotiated reimbursement prices of anticancer drugs from relevant official documents issued by the MOHRSS, and the NHSA, and searched the Drugdataexpy when prices were not publicly available [27]. Pre-negotiation prices for these anticancer drugs closest to the time of negotiation were collected from the Drugdataexpy [27]. The treatment costs for each therapeutic indication over an expected treatment duration were estimated using dosing information from drug labels, as well as both pre-negotiation prices and negotiated reimbursement prices. This approach accounts for differences in the duration of treatment across anticancer drugs and therapeutic indications [32]. The median treatment duration for each indication was collected from the pivotal trial, representing the expected treatment duration when calculating treatment costs [14]. For indications for which dosages depended on body surface area or weight, we assumed a patient weighing 60 kg with a body surface area of 1.6 m$^2$ in consistent with NHSA requirements for dossiers of drugs to be negotiated. We converted treatment costs to US dollars by applying the exchange rates for the respective years of negotiation, obtaining from the OECD.Stat: Organisation for Economic Co-operation and Development's statistical database.

We also collected data on whether the anticancer drug was domestically developed, whether the therapeutic indication was approved through priority review or conditional approval, as well as the cancer site, the baseline survival, the line of therapy, the comparator, the blind method, and the administration route as control variables [19]. A list of the indications included in the study sample and an overview of all variables including control variables are in S1–S3 Tables.

## Statistical analysis

Treatment costs over expected treatment durations for therapeutic indications were calculated [14], and median costs before price negotiation and after price negotiation were compared. We identified the cost difference between indications supported by randomized controlled

trials and indications supported by single-arm clinical trials. Mann–Whitney U tests were utilized for assessing differences between groups.

We employed regression analyses to evaluate the association between treatment costs and clinical value, both before and after negotiation. To examine the impact of price negotiation on this association, we then considered interaction effects between clinical value and the negotiation status in the multivariate models (model specification can be found in S2 Text). We distinguished between indications supported by randomized controlled trials and indications supported by single-arm clinical trials when conducting data analysis, given their differences in trial design and in the measurement of clinical value. Because our dependent variables all had skewed distributions, we applied log-linear regressions. We included control variables sequentially and separately in each model to avoid potential overfitting, following the method employed by Howard and colleagues [19,32]. To ensure the robustness of our findings, we performed several sensitivity checks, as elaborated in the S3 Text.

We examined whether price negotiation has led to a reduction in the variation of treatment costs for a specific value. As data were sparse for certain values, we employed a model-based approach to estimate the price variance for a given value, denoted as $Var(Y|X) = E(Y^2|X) - E(Y|X)^2$, where $Y$ represents treatment costs and $X$ represents clinical value. Specifically, we developed 2 ordinary least squares regression models, one for modeling $Y$ as a function of $X$ and another for modeling $Y^2$ as a function of $X$, respectively. We, therefore, obtained the estimates of $E(Y^2|X)$ and $E(Y|X)^2$ using the regression coefficients.

All data were collected using a predesigned Excel file and were imported into R (version 4.1.0) for statistical analysis. The ggplot2 (version 3.3.5) was used for visualization. All statistical tests were two-sided, and statistical significance was defined as a two-sided $P$ value of less than 0.05.

## Results

Our final sample included 103 indications that were negotiated from 2017 to 2022, 76 indications of which were supported by randomized controlled trials and 27 indications of which were supported by single-arm clinical trials (Fig 1). The basic characteristics of these indications were shown in Table 1.

### Pre-negotiation prices and price changes after negotiation

Before price negotiation, the median treatment costs over expected durations of indications in the entire sample were US\$34,460.72 (interquartile range (IQR): 19,990.49 to 55,441.66), which significantly decreased to US\$13,688.79 (IQR: 7,746.97 to 21,750.97) after price negotiation went into effect ($P < 0.001$) (Fig 2). The median difference between treatment costs before negotiation versus after negotiation was US\$18,499.43 (IQR: 10,449.52 to 36,921.74), representing a relative reduction in treatment costs over the period of negotiation of 60%. The median treatment costs of indications supported by randomized controlled trials were not significantly different from those supported by single-arm clinical trials, neither before price negotiation [US\$33,894.51 (IQR: 17,783.96 to 59,187.64) versus US\$37,696.18 (IQR: 25,150.66 to 51,985.30), $P = 0.634$] nor after price negotiation [US\$13,819.43 (IQR: 6,743.64 to 21,750.97) versus US\$12,504.74 (9,665.06 to 21,230.96), $P = 0.913$] (Fig 2).

### Changes in associations between prices and clinical value after price negotiation

For 76 indications supported by randomized controlled trials, the median survival benefits in either OS or PFS were 4.9 months (IQR: 2.4 to 8.5) compared to their reference drugs. Among

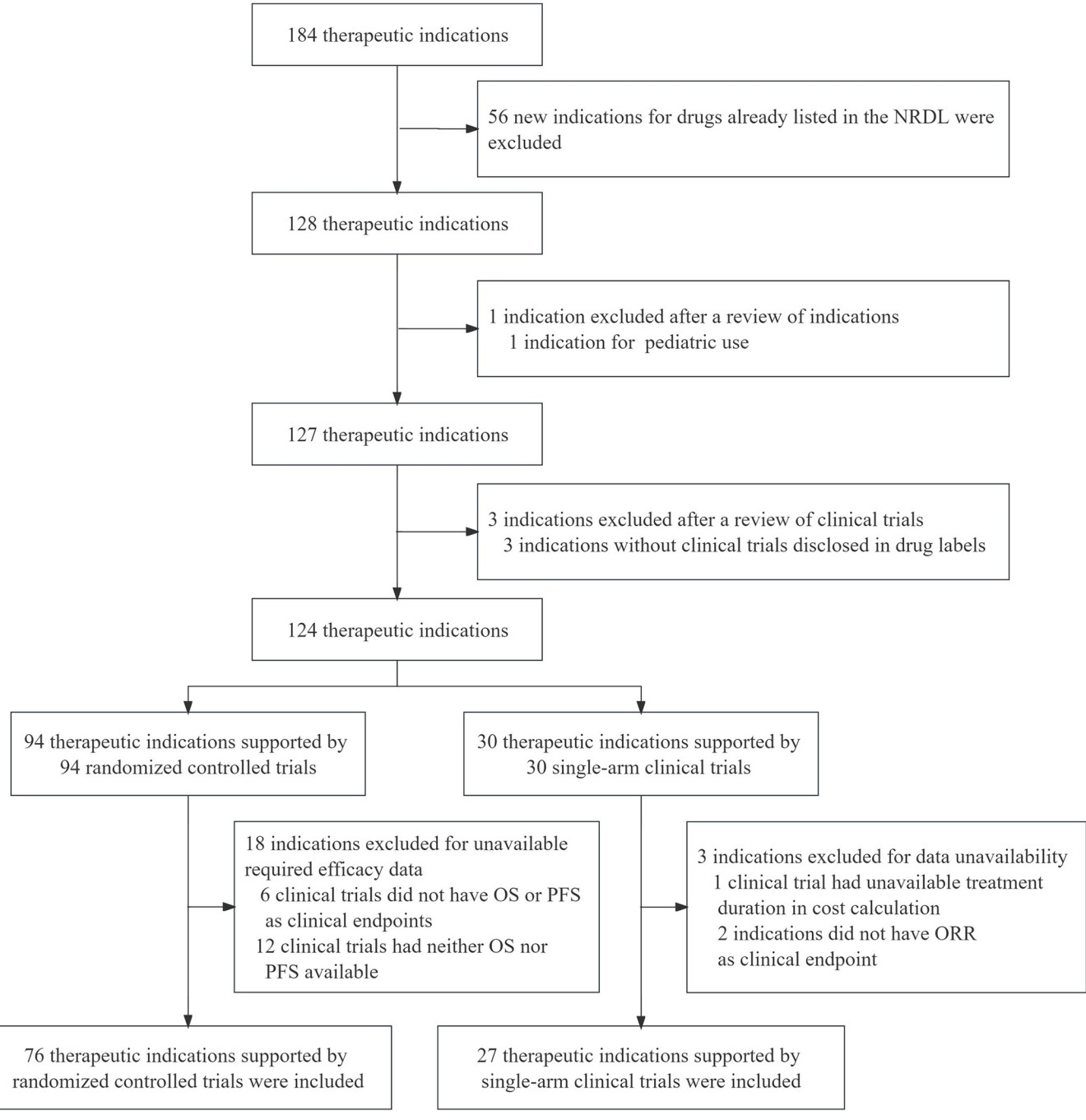

**Fig 1. Flowchart of sample selection.** NRDL, National Reimbursement Drug Lists; ORR, overall response rate; OS, overall survival; PFS, progression-free survival.

these, 27 exhibited no difference, 21 demonstrated improvement, and 28 showed a reduction or had unavailable data in QoL. Sixty-two indications experienced a reduction in safety, while 14 indicated safety improvement or no difference. Comparing the treatment costs of these indications with their survival benefits, we observed a positive unadjusted association both before and after negotiation (Fig 3A). After excluding the extreme outliers and log-

**Table 1. Characteristics of indications in the study sample from China.**

| Indications supported by randomized controlled trials (*N* = 76) | Number | Percent |
|---|---|---|
| Cancer site | | |
| Blood | 15 | 20 |
| Lung | 15 | 20 |
| Breast | 11 | 14 |
| Colorectal | 5 | 7 |
| Renal | 5 | 7 |
| Other | 25 | 33 |
| Domestically developed | | |
| Yes | 19 | 25 |
| No | 57 | 75 |
| Priority review | | |
| Yes | 37 | 49 |
| No | 39 | 51 |
| Administration route | | |
| Intravenous | 23 | 30 |
| Oral | 53 | 70 |
| First-line therapy | | |
| Yes | 30 | 39 |
| No | 46 | 61 |
| Blind | | |
| Yes | 38 | 50 |
| No | 38 | 50 |
| Comparator | | |
| Placebo | 25 | 33 |
| Active | 51 | 67 |
| Year of approval | | |
| Before 2017 | 29 | 38 |
| 2017 and beyond | 47 | 62 |
| **Indications supported by single-arm clinical trials (*N* = 27)** | **Number** | **Percent** |
| Cancer site | | |
| Hematological | 14 | 52 |
| Non-hematological | 13 | 48 |
| Domestically developed | | |
| Yes | 18 | 67 |
| No | 9 | 33 |
| Conditional approval | | |
| Yes | 19 | 70 |
| No | 8 | 30 |
| Administration route | | |
| Intravenous | 13 | 48 |
| Oral | 14 | 52 |
| First-line therapy | | |
| Yes | 4 | 15 |
| No | 23 | 85 |
| Year of approval | | |
| Before 2017 | 4 | 15 |
| 2017 and beyond | 23 | 85 |

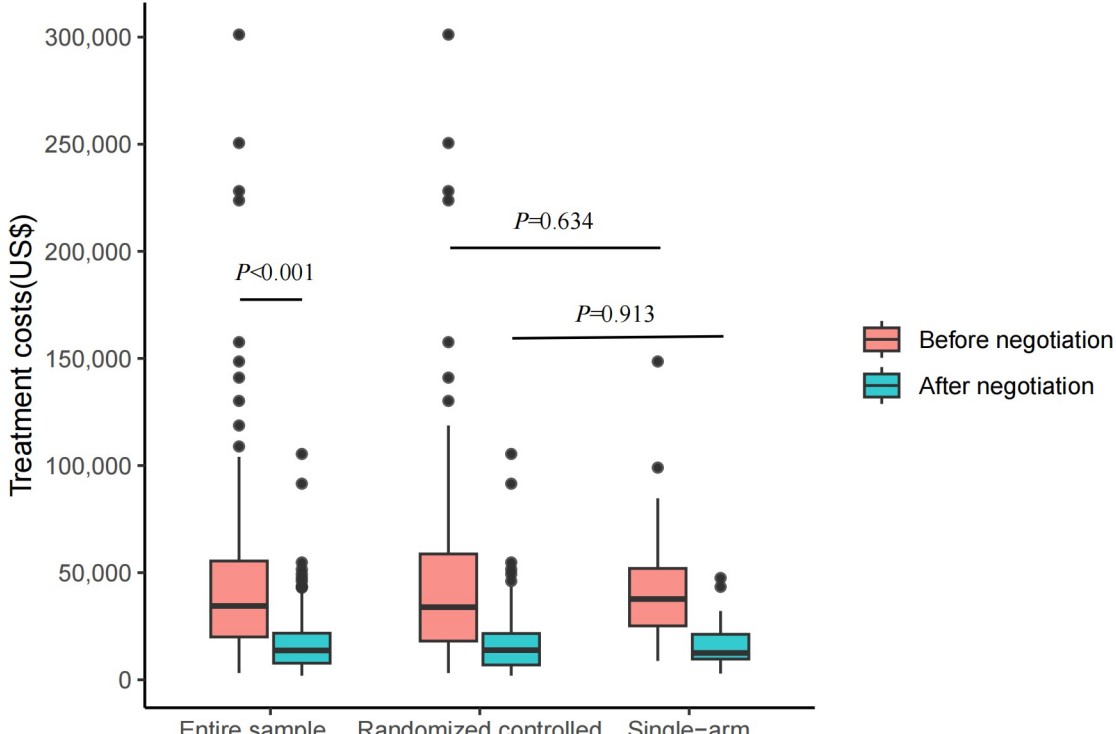

**Fig 2. Treatment costs over expected durations for indications both before and after negotiation in China.** This boxplot shows costs over expected treatment durations of indications over the entire sample (*N* = 103), of indications supported by randomized controlled trials (*N* = 76), and of indications supported by single-arm clinical trials (*N* = 27), both before and after negotiation. Entire sample = indications supported by both randomized controlled trials and single-arm clinical trials; randomized controlled = indications supported by randomized controlled trials; and single-arm = indications supported by single-arm clinical trials. Pink color represents indications before negotiation, while blue-green color represents indications after negotiation. Asymptotic *P* values were attached to show group differences. The box displays the median and IQR. The band near the middle of the box is the median, and the bottom and top of the box are the first and third quartiles (the 25th and 75th percentiles, respectively). The solid lines below and above the box describe the bottom and top whiskers. The small dots indicate extreme outliers. IQR, interquartile range.

transforming treatment costs based on data distribution, the positive adjusted associations between treatment costs and survival benefits were also observed (Fig 3B).

Including measures of clinical value on survival, QoL, and safety in multiple regression analysis revealed that each additional month of survival gained was associated with an average increase in treatment costs of 3.4% (95% confidence interval (CI) [2.1, 4.8], *P* < 0.001) before price negotiation and 3.2% (95% CI [1.9, 4.4], *P* < 0.001) after negotiation, respectively [Table 2, Model (1) and Model (2)]. By contrast, neither comparative safety nor QoL was associated with treatment costs before or after negotiation. In the interaction model [Table 2, Model (3)], the estimates of the interaction terms between price negotiation and added survival, safety, and QoL on treatment costs were all negative and insignificant, suggesting that price negotiation may not have significantly changed the association of treatment costs with clinical value.

For indications supported by single-arm clinical trials, the median ORR was 0.69 (IQR: 0.36 to 0.77). We observed a positive unadjusted association between treatment costs and ORR both before and after negotiation (Fig 4A). After log-transforming treatment costs based on data distribution, the adjusted positive associations were also observed (Fig 4B). Regression analyses indicated that a 10% increase in ORR was associated with a 6.0% (95% CI [1.6, 10.3],

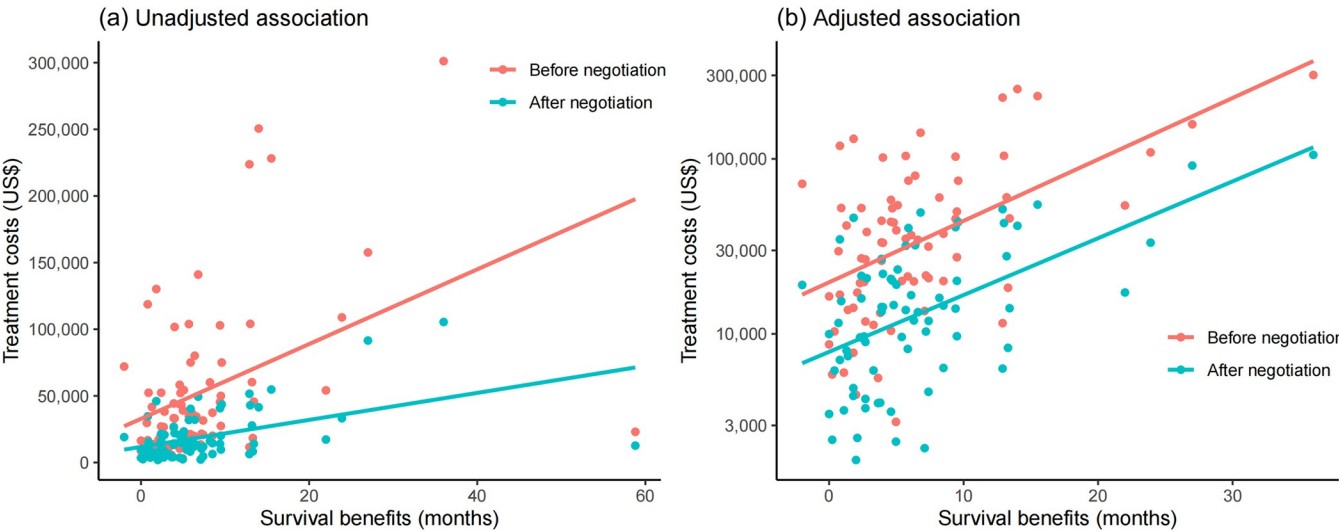

**Fig 3. Associations between treatment costs and survival benefits for indications supported by randomized controlled trials.** Fig 3A displays the raw data of 76 indications supported by randomized controlled trials, illustrating the unadjusted association between treatment costs and survival benefits. Fig 3B depicts the adjusted association between treatment costs and survival benefits for 75 indications after excluding the extreme outliers and log-transforming treatment costs based on data distribution. Notably, the Y-axis scale for treatment costs in Fig 3B has been log-transformed, while the axis labels display the original values for a clearer visual representation. Pink dots represent indications before negotiation, while blue-green dots represent indications after negotiation. Lines indicate the associations between treatment costs and survival benefits, with a pink line for the association before negotiation and a blue line for the association after negotiation.

*P* = 0.009) increase in treatment costs before price negotiation and a 6.9% (95% CI [1.8, 12.0], *P* = 0.010) increase after price negotiation [Table 3, Model (1) and Model (2)]. In the interaction model with added interaction terms of price negotiation and ORR, the estimate was positive but insignificant, implying that the relationship between treatment costs and ORR may not be mediated by price negotiation [Table 3, Model (3)].

The results of the above models were largely robust to the inclusion of control variables (S4–S9 Tables). Re-estimating regression models for indications supported by randomized controlled trials using survival benefits in OS and survival benefits in PFS as the survival measure, respectively, instead of aggregated survival benefits in either OS or PFS, yielded highly consistent results. Furthermore, our findings remained robust when we excluded anticancer drugs launched before the implementation of price negotiation (2017) in our sample.

### Variations of prices for a given value after price negotiation

For indications supported by randomized controlled trials with survival benefits in OS or PFS, the implementation of price negotiation has resulted in a reduction in the variation of drug treatment costs for a given value, with greater reductions observed for higher values (Fig 5A). Similar results were observed for indications supported by single-arm clinical trials (Fig 5B). In other words, for drugs with the same clinical value, the treatment costs have become more centralized after price negotiation.

### Discussion

In our study, we found that while the implementation of price negotiation has reduced drug prices significantly, the positive associations between prices and the clinical value of anticancer drugs, identified before negotiation, remained largely unchanged after negotiation. Moreover,

**Table 2. Associations between treatment costs and clinical value for indications supported by randomized controlled trials in China.**

| Variables | Model (1) Costs before negotiation | | Model (2) Costs after negotiation | | Model (3) Costs before and after negotiation | |
|---|---|---|---|---|---|---|
| | Coefficient (95% CI) | *P* value | Coefficient (95% CI) | *P* value | Coefficient (95% CI) | *P* value |
| Survival benefits in months | 0.034 (0.021, 0.048) | <0.001 | 0.032 (0.019, 0.044) | <0.001 | 0.034 (0.022, 0.047) | <0.001 |
| QoL (ref = No difference) | | | | | | |
| Improvement | −0.070 (−0.289, 0.149) | 0.526 | −0.102 (−0.305, 0.100) | 0.317 | −0.070 (−0.279, 0.139) | 0.510 |
| Reduction or unavailability | −0.145 (−0.349, 0.058) | 0.158 | −0.167 (−0.355, 0.022) | 0.082 | −0.145 (−0.340, 0.049) | 0.141 |
| Safety (ref = Reduction) | | | | | | |
| Improvement or no difference | 0.134 (−0.089, 0.357) | 0.235 | 0.083 (−0.124, 0.289) | 0.428 | 0.134 (−0.079, 0.347) | 0.216 |
| Negotiation (ref = Before negotiation) | | | | | | |
| After negotiation | | | | | −0.369 (−0.602, −0.137) | 0.002 |
| Negotiation × Survival benefits (ref = Before negotiation × Survival benefits) | | | | | | |
| After negotiation × Survival benefits | | | | | −0.003 (−0.021, 0.015) | 0.751 |
| Negotiation × QoL (ref = Before negotiation × No difference) | | | | | | |
| After negotiation × Improvement | | | | | −0.033 (−0.328, 0.263) | 0.828 |
| After negotiation × Reduction or unavailability | | | | | −0.021 (−0.296, 0.254) | 0.879 |
| Negotiation × Safety (ref = Before negotiation × Reduction) | | | | | | |
| After negotiation × Improvement or no difference | | | | | −0.051 (−0.353, 0.250) | 0.737 |

We log-transformed treatment costs for these regression analyses. CI, confidence interval; QoL, quality of life. Negotiation × Clinical value (survival benefits, QoL, or safety) refers to the interaction term involving negotiation status and measures of clinical value. Of note, because associations were strongly influenced by the outlier(s), the Rituximab for the treatment of diffuse large-B-cell lymphoma, we excluded the outlier(s) from these analyses.

the variation of drug treatment costs for a specific value decreased following price negotiation, with a more substantial decrease observed for higher clinical values.

In the entire sample of anticancer drug indications, price reduction resulted in a median treatment cost reduction of $18,499.43 (60%). This reduction highlights the significant bargaining power of Chinese health authorities, possibly stemming from the largest population of China in the world. The achievement has contributed to the improvement of affordability, accessibility, and utilization of anticancer drugs in China [15–18]. There was no significant difference in treatment costs between indications supported by randomized controlled trials and those indications supported by single-arm clinical trials both before and after price negotiation, implicitly indicating that the strength of clinical evidence may not strongly influence the decision-making in the price negotiation process for anticancer drugs addressing unmet clinical needs. Indeed, most of the indications supported by single-arm clinical trials were approved through conditional approval, which was specialized for the approval of health technology treating serious and life-threatening diseases without available effective treatments. This finding was consistent with the previous study, which concluded that Chinese health authorities have placed a high priority on meeting clinical needs and addressing therapeutic gaps in the NRDL while reducing the requirements for clinical trials [33]. Nevertheless, extra

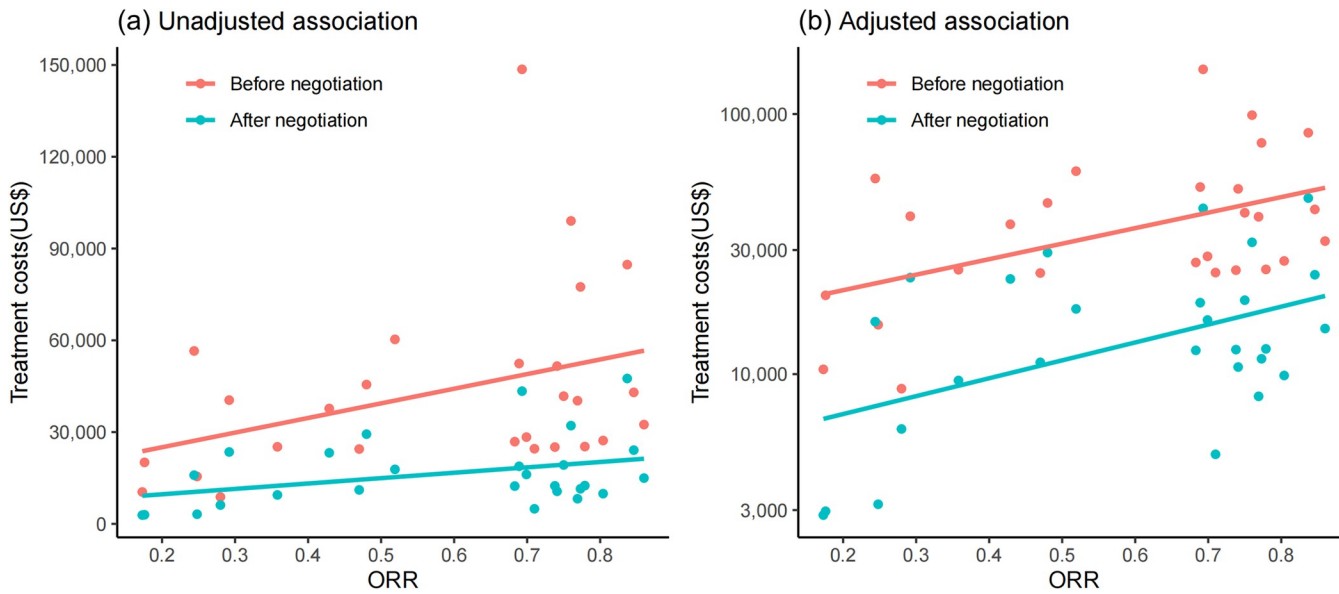

**Fig 4. Associations between treatment costs and overall response rates for indications supported by single-arm clinical trials.** ORR, overall response rates. Fig 4A displays the raw data of 27 included indications supported by single-arm clinical trials, illustrating the unadjusted association between treatment costs and ORR. Fig 4B depicts the adjusted association between treatment costs and ORR after log-transforming treatment costs based on data distribution. Notably, the Y-axis scale for treatment costs in Fig 4B has been log-transformed, while the axis labels display the original values for a clearer visual representation. Pink dots represent indications before negotiation, while blue-green dots represent indications after negotiation. Lines indicate the association between treatment costs and ORR, pink line for the association before negotiation and blue line for the association after negotiation.

attention could be paid to addressing the uncertainty of clinical evidence or accounting for the strength of clinical evidence when setting pricing strategies in China.

In our study, we found that the added survival in either OS or PFS of indications supported by randomized controlled trials and ORR of indications supported by single-arm clinical trials were positively associated with treatment costs, both before and after negotiation. In other words, not only have Chinese health authorities been implementing value-based pricing, but pharmaceutical companies may also be using this pricing strategy when setting market prices [34]. The regression results showed that the association of prices with the clinical value of anticancer drugs may not have been affected by price negotiation, indicating that the value increases of anticancer drugs could still be reflected in the magnitude of costs after substantial price reductions. Furthermore, price negotiation has decreased the variation of drug prices for a given value, with higher reductions for higher values. This means the implementation of

**Table 3. Associations between treatment costs and clinical value for indications supported by single-arm clinical trials in China.**

| Variables | Model (1) Costs before negotiation | | Model (2) Costs after negotiation | | Model (3) Costs before and after negotiation | |
|---|---|---|---|---|---|---|
| | Coefficient (95% CI) | P value | Coefficient (95% CI) | P value | Coefficient (95% CI) | P value |
| ORR | 0.597 (0.163, 1.030) | 0.009 | 0.688 (0.179, 1.197) | 0.010 | 0.596 (0.140, 1.051) | 0.011 |
| Negotiation (ref = Before negotiation) | | | | | | |
| After negotiation | | | | | −0.495 (−0.899, −0.090) | 0.018 |
| Negotiation × ORR (ref = Before negotiation × ORR) | | | | | | |
| After negotiation × ORR | | | | | 0.092 (−0.552, 0.735) | 0.776 |

We log-transformed treatment costs for these regression analyses. CI, confidence interval; ORR, overall response rate. Negotiation × ORR refers to the interaction term involving negotiation status and ORR.

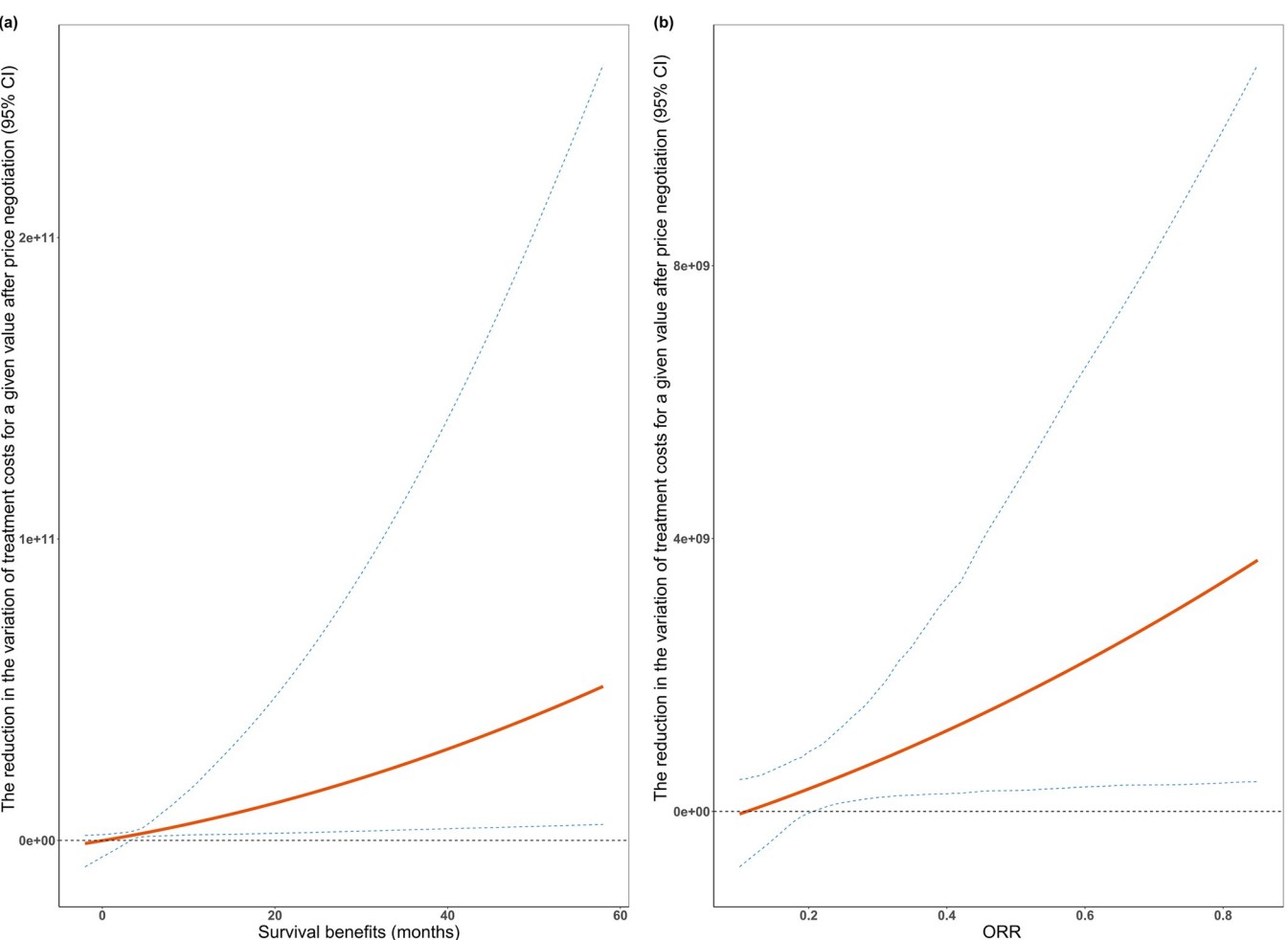

**Fig 5. Reductions in the variation of treatment costs for a given value after price negotiation.** ORR, overall response rates. Fig 5a shows the survival benefits and the reduction in the variation of treatment costs for a given survival after price negotiation for indications supported by randomized controlled trials. Fig 5b shows the ORR and the reduction in the variation of treatment costs for a given ORR after price negotiation for indications supported by single-arm clinical trials. The red line represents the reduction in the variation of treatment costs after price negotiation for a given value, and the blue dotted lines represent the 95% CI.

price negotiation in China has led to drug pricing being better aligned with clinical value, especially for high-value anticancer drugs.

The alignment of drug prices and value has great potential to benefit patients and health systems from 2 interrelated dimensions: accounting for the value of drugs and optimizing medical resource allocations [35]. The economic value of anticancer drugs should reflect the magnitude of health gain to justify price increases, and funds allocated to lower-value anticancer drugs should be redirected toward more valuable treatments to maximize health outcomes. For instance, drugs whose prices do not match their value are supposed to be subjected to lower prices during the negotiation process, which enables finite resources to be allocated towards treatments that offer patients greater clinical benefits. The accomplishments in China in the regard hold substantial potential for offering valuable insights into drug price regulation, not only for other low- and middle-income countries grappling with resource constraints and escalating drug expenditures but also for high-income nations. The fact that high-expenditure drugs will be subjected to price negotiation in the United States under the Inflation Reduction

Act of 2022 highlights the increasing recognition of the significant influence that price negotiation can exert. In this context, China serves as an example of how price negotiation can be designed to better align prices with clinical value in addition to reducing drug prices.

Existing studies on the association between costs and clinical value of anticancer drugs have predominantly been conducted in the US and Europe, with a few studies in Japan and China, most of which demonstrated only weak or no association between prices and value [14,30,36–38]. However, sample and methodological differences across studies have hindered comparability [38]. In particular, the primary differences related to the calculation of treatment costs may explain different conclusions observed among studies and countries [38]. Additionally, the lack of international comparisons involving China using the same methodology and the same sample makes it difficult to assess the strength of the association estimates found in our findings sufficiently, future research is warranted to address this gap. Moreover, the impact of pricing-related policy on the association of costs and clinical value was under-researched. One study within the context of Germany revealed that price regulation in Germany had better aligned prices with clinical benefit, which was in line with our findings [19]. Further research for other countries in this regard is encouraged.

To our knowledge, our study is the first to evaluate the impact of price negotiation on the relationship between prices and the clinical value of anticancer drugs in China. We tried to address several issues after benefiting from existing studies. Firstly, treatment duration varies across anticancer drugs, making daily cost and monthly cost imprecise measures of the total financial impact from treatment and biasing price and value comparisons [14]. Therefore, treatment durations over expected durations were considered in our study when calculating costs. Secondly, we included clinical measures (i.e., safety and QoL) that matter to patients other than survival. Finally, simple correlation studies may not adequately control for confounding variables that can affect the relationship between prices and value, thus multiple regression analyses were used in this study [19].

This study has several limitations. First, while China has been increasing the transparency in the decision-making process of price negotiation, the availability of relevant data has been limited to varying degrees, particularly in previous years. Therefore, our study relied on data obtained from other sources, which may not be fully representative of the information reviewed by health authorities. Second, in cases where OS data in median times was not available, we used added survival in PFS as a surrogate. Although PFS and OS are considered primary clinical endpoints in China, OS is generally preferred when both OS and PFS are available [39]. In addition, one of the categories of QoL was "reduction or unavailability" due to the low rate of QoL reporting. Evidence showed that the reporting of QoL in clinical trials was associated with positive trial outcomes, while harm was underreported in clinical trials [40–42]. Third, we only included measures of clinical value in our main analysis. As drug evaluation continues to evolve towards comprehensive value assessment, subsequent price negotiations in China have incorporated measures reflecting other aspects of value, such as equity and innovation. Nevertheless, measures of clinical value are the core elements considered by health authorities and have been consistently assessed every year. Finally, the sample size was modest for indications supported by randomized controlled trials (76 indications) and was limited for indications supported by single-arm clinical trials (27 indications), preventing us from including all potentially relevant control variables simultaneously in regression analyses and restricting the ability to detect associations between variables. Future studies could build on this analysis by re-running it to include additional anticancer drugs that will be successfully negotiated in the future.

In conclusion, the price negotiation implemented in China in recent years has significantly reduced the prices of anticancer drugs, while the positive association between prices and

clinical value has remained largely unchanged. Importantly, price negotiation has led to drug pricing being better aligned with clinical value. China's achievements have great implications for price regulation in other countries facing rising drug expenditures and constrained resources.

## Supporting information

**S1 Text. Price negotiation in China.**
(DOCX)

**S2 Text. Regression equations of models.**
(DOCX)

**S3 Text. Sensitivity analyses.**
(DOCX)

**S1 Table. Overview of variables.**
(DOCX)

**S2 Table. Information on 76 indications of anticancer drugs supported by randomized controlled trials in China.**
(DOCX)

**S3 Table. Information on 27 indications of anticancer drugs supported by single-arm clinical trials in China.**
(DOCX)

**S4 Table. Associations between treatment costs and clinical value, including control variables, for indications supported by randomized controlled trials before price negotiation in China.**
(DOCX)

**S5 Table. Associations between treatment costs and clinical value, including control variables, for indications supported by randomized controlled trials after price negotiation in China.**
(DOCX)

**S6 Table. Associations between treatment costs and clinical value, including interaction terms and control variables, for indications supported by randomized controlled trials in China.**
(DOCX)

**S7 Table. Associations between treatment costs and ORR, including control variables, for indications supported by single-arm clinical trials before price negotiation in China.**
(DOCX)

**S8 Table. Associations between treatment costs and ORR, including control variables, for indications supported by single-arm clinical trials after price negotiation in China.**
(DOCX)

**S9 Table. Associations between treatment costs and ORR, including interaction terms and control variables, for indications supported by single-arm clinical trials in China.**
(DOCX)

## Author Contributions

**Conceptualization:** Jing Zhou, Jay Pan.

**Data curation:** Jing Zhou, Hao Lu.

**Formal analysis:** Jing Zhou, Tianjiao Lan, Hao Lu, Jay Pan.

**Funding acquisition:** Jay Pan.

**Investigation:** Jing Zhou, Hao Lu.

**Methodology:** Jing Zhou, Tianjiao Lan, Hao Lu, Jay Pan.

**Resources:** Jay Pan.

**Software:** Jing Zhou, Tianjiao Lan, Hao Lu.

**Validation:** Jing Zhou.

**Visualization:** Jing Zhou, Tianjiao Lan, Hao Lu.

**Writing – original draft:** Jing Zhou.

**Writing – review & editing:** Tianjiao Lan, Hao Lu, Jay Pan.

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
