## [Editor Report · Decision Letter 0]

19 Oct 2023

Dear Dr Pan, 

Thank you for submitting your manuscript entitled "Impact of price negotiation on the pricing of anticancer drugs in China" for consideration by PLOS Medicine.

Your manuscript has now been evaluated by the PLOS Medicine editorial staff and I am writing to let you know that we would like to send your submission out for external peer review.

Please re-submit your manuscript within two working days, i.e. by Oct 23 2023 11:59PM.

Feel free to email me if you have any queries relating to your submission.

Kind regards,

Philippa Dodd, MBBS MRCP PhD

PLOS Medicine

pdodd@plos.org

---

## [Decision Letter · Decision Letter 1]

22 Nov 2023

Dear Dr. Pan,

Thank you very much for submitting your manuscript "Impact of price negotiation on the pricing of anticancer drugs in China" (PMEDICINE-D-23-02974R1) for consideration at PLOS Medicine. 

[LINK]

In light of these reviews, I am pleased to say that we would like to consider a revised version that addresses the reviewers' and editors' comments. Obviously we cannot make any decision about publication until we have seen the revised manuscript and your response, and we plan to seek re-review by one or more of the reviewers. 

We expect to receive your revised manuscript by Dec 13 2023 11:59PM. Please email us (plosmedicine@plos.org) if you have any questions or concerns.

We look forward to receiving your revised manuscript. 

Best wishes,

Pippa

Philippa Dodd, MBBS MRCP PhD

PLOS Medicine

plosmedicine.org

pdodd@plos.org

COMMENTS FROM THE ACADEMIC EDITOR:

For me, Tables 2 and 3 were very hard to follow -- they seem like the kind of tables that belong in an economics journal, not PLOS. 

Why do Figs 3 and 4 highlight the unadjusted association -- those Figures make it seem like a major change in the association after negotiation, but then somehow it ends up being between 3.2 and 3.4 percent increase per life-month gained? ("Life-months gained", btw, is not a good phrase to use when talking about PFS, only OS.)

I also couldn't follow the y axis on Fig 5. I also think the authors should report the general statistics on OS/PFS/QoL or ORR in the Results before showing the changes over time, so that readers can put the changes in context. 

I also had some trouble with the Discussion -- how does this study provide guidance for other countries negotiating prices? The authors don't describe much on how negotiation occurred, only that it did occur and the outcomes of it. Can the authors estimate how much savings to the health care system overall were achieved given the spending on the drugs in the study?

Also, aren't there some potential concerns about the finding that there's greater reduction in prices for drugs offering the most clinical benefits? That seems like it works against private profit incentives to generate those most-valuable drugs.

COMMENTS FROM THE EDITORS:

GENERAL

Thank you for submitting this version of your manuscript which considers previous comments. We were grateful for your detailed and considered responses to previous reviewer comments, which we accept.

Please respond to all additional comments detailed below.

TITLE

Please revise your title according to PLOS Medicine's style. Your title must be nondeclarative and not a question. It should begin with main concept if possible. "Effect of" should be used only if causality can be inferred, i.e., for an RCT. Please place the study design ("A randomized controlled trial," "An observational study," "A modelling study," etc.) in the subtitle (i.e., after a colon).

ABSTRACT

Please structure your abstract using the PLOS Medicine headings (Background, Methods and Findings, Conclusions).

Please combine the Methods and Findings sections into one section, “Methods and findings”.

Abstract Background: Provide the context of why the study is important. The final sentence should clearly state the study question.

Abstract Methods and Findings:

Please ensure that all numbers presented in the abstract are present and identical to numbers presented in the main manuscript text.

Please include the study design, population and setting, the total number of indications you report on (103), and please explicitly state the main outcome measures.

Please include (brief) details of where and how you extracted relevant data to inform your study.

It might be helpful to include examples of the indications that you refer to and perhaps the different drug classes.

Please quantify the main results with 95% CIs as well as p values. When reporting p values please report as <0.001 and where higher as p=0.002, for example. When reporting 95% CIs please separate upper and lower bounds with commas instead of hyphens as the latter can be confused with reporting of negative values.

Please include any important dependent variables that are adjusted for in the analyses.

In the last sentence of the Abstract Methods and Findings section, please describe the main limitation(s) of the study's methodology.

Abstract Conclusions:

The opening sentence is currently very long and not very accessible (line 47, in particular). As written, it’s difficult to fully appreciate the implications of your study. Please revise in mind of the below points:

* Please address the study implications without overreaching what can be concluded from the data; the phrase "In this study, we observed ..." may be useful.

* Please interpret the study based on the results presented in the abstract, emphasizing what is new without overstating your conclusions.

* Please avoid vague statements such as "these results have major implications for policy/clinical care". Mention only specific implications substantiated by the results.

* Please avoid assertions of primacy ("We report for the first time....")

AUTHOR SUMMARY

At this stage, we ask that you include a short, non-technical Author Summary of your research to make findings accessible to a wide audience that includes both scientists and non-scientists. The authors summary should consist of 2-3 succinct bullet points under each of the following headings:

• Why Was This Study Done? Authors should reflect on what was known about the topic before the research was published and why the research was needed.

• What Did the Researchers Do and Find? Authors should briefly describe the study design that was used and the study’s major findings. Do include the headline numbers from the study, such as the sample size and key findings. 

• What Do These Findings Mean? Authors should reflect on the new knowledge generated by the research and the implications for practice, research, policy, or public health. Authors should also consider how the interpretation of the study’s findings may be affected by the study limitations. In the final bullet point of ‘What Do These Findings Mean?’, please describe the main limitations of the study in non-technical language.

The Author Summary should immediately follow the Abstract in your revised manuscript. This text is subject to editorial change and should be distinct from the scientific abstract. Please see our author guidelines for more information: https://journals.plos.org/plosmedicine/s/revising-your-manuscript#loc-author-summary

INTRODUCTION

If there has been a systematic review of the evidence related to your study (or you have conducted one), please refer to and reference that review and indicate whether it supports the need for your study.

Line 104 – the term ‘more aligned’ is a little vague please revise for clarity.

METHODS and RESULTS

Line 110 – please (re)define NDRL here for the reader.

Please clearly justify the start date of 2017 in the methods section. Throughout, please ensure all abbreviations are defined at first use for the reader.

Please ensure that outcome measures are quantified with 95% CIs and p values. When reporting p values please report as <0.001 and where higher the exact p values a 0.002, for example.

In the manuscript text, please ensure that you indicate: 

(1) the specific hypotheses you intended to test, 

(2) the analytical methods by which you planned to test them, 

(3) the analyses you actually performed, and 

(4) when reported analyses differ from those that were planned, transparent explanations for differences that affect the reliability of the study's results. If a reported analysis was performed based on an interesting but unanticipated pattern in the data, please be clear that the analysis was data-driven.

TABLES

Please ensure that titles and captions are supplied for all tables (including those in Supporting Information files) which clearly describe the table content without the need to refer to the text. 

Please ensure that all abbreviations including those used for statistical reporting are defined the caption or an appropriate footnote.

Throughout, please indicate whether your analyses are adjusted or unadjusted and where you present adjusted analyses, please clearly detail the factors which have been adjusted for. To help facilitate transparent data reporting please also present the unadjusted analyses for comparison.

Throughout, when reporting outcome measures please quantify with 95% CIs and p values. Please report as <0.001 and where higher the exact p value as 0.002, for example. Please avoid the use of asterisks to denote significance levels. Please use commas to separate upper and lower bounds of CIs.

Please see the academic editor comments above regarding your tables which we agree with. Please revise for improved clarity as requested.

Table 1 – what does ‘origin country’ refer to in this context?

Table 2 & 3 – we agree that these are hard to follow, the column headers are also uninformative. Please revise in mind of the aforementioned comments.

FIGURES

Please see the academic editor comments (above) regarding your figures which we agree with. Please revise as requested.

Please ensure that titles and captions are supplied for all figures (including those in Supporting Information files). 

Please ensure that all abbreviations including those used for statistical reporting are defined the caption or an appropriate footnote.

Throughout, please quantify main outcome measures with 95% CIs and p values. Please report p as <0.001 and where higher the exact p value as 0.002, for example. Please use commas as opposed to hyphens to separate upper and lower CI bounds as the latter can be confused with reporting of negative values.

To help facilitate transparent data reporting please justify any deviations from the above.

Figures 3 and 4 – please present the adjusted results. For the purpose of transparent data reporting the unadjusted results can be presented for comparison.

Figure 5 – we agree that the axis and labels are confusing, please revise

DISCUSSION

Please also see the comments from the academic editor (above) regarding your discussion which we agree with. Please include additional discussion of these specific points.

Opening paragraph, line 284 onwards – as for the abstract conclusions this is a little difficult to follow and for the more general reader could be particularly difficult. Suggest revising for improved clarity/accessibility. 

Please present and organize the Discussion as follows: a short, clear summary of the article's findings; what the study adds to existing research and where and why the results may differ from previous research; strengths and limitations of the study; implications and next steps for research, clinical practice, and/or public policy; one-paragraph conclusion. Please refrain from the use of sub-headings such that the discussion reads as continuous prose.

REFERENCES

For in-text reference callouts please place citations in square parentheses separate by commas. For example, [1,3,6] or [1-3]. Please check and amend throughout all sub-sections of the manuscript and supporting files.

In the bibliography, please ensure that you list up to but no more than 6 author names followed by et al.

For all web references please ensure you include an, ‘Accessed [date].’

Journal name abbreviations should be those listed in the National Center for Biotechnology Information (NCBI) databases.

SUPPORTING INFORMATION

Please cite your Supporting Information as outlined here: https://journals.plos.org/plosmedicine/s/supporting-information

In the published article, supporting information files are accessed only through a hyperlink attached to the captions. For this reason, you must list captions at the end of your manuscript file. You may include a caption within the supporting information file itself, as long as that caption is also provided in the manuscript file. Do not submit a separate caption file.

Tables S4-S9 – these are similarly difficult to follow as those in the main manuscript. Please revise in mind of the comments detailed above and to ensure consistency of reporting throughout your manuscript.

COMMENTS FROM THE REVIEWERS:

Reviewer #1: In general, the authors have thoughtfully addressed the majority of my comments. The manuscript has seen significant improvement. I have one minor suggestion related to my third comment, which the authors may want to consider: 

While some products have entered medical insurance negotiations in China based on conditional approval, we should still provide some discussion on this matter. It is advisable to suggest that the national medical insurance department should have distinct pricing policies for products that have not yet demonstrated clinical benefits. Alternatively, we can explore how other countries handle medical insurance policies for conditionally approved products to consider how we should proceed.

Reviewer #2: The authors had explained all the questions reasonably and revised the corresponding sections. The manuscript is proposed for publication.

Reviewer #3: This study assessed the impact of China's annual price negotiations on anticancer drugs, implemented since 2017, focusing on their cost and clinical value. It included indications of anticancer drugs negotiated from 2017 to 2022, analyzing clinical value data from pivotal trials (covering safety, survival, quality of life, and response rates) and calculating treatment costs. Using multivariate regression, the study examined the relationship between drug costs and clinical value pre- and post-negotiation. Results revealed a significant reduction in median treatment costs after negotiations, from $34,460.72 to $13,688.79. Initially, higher treatment costs correlated with greater clinical value. Post-negotiation, this correlation was less significant, though the variation in costs for given clinical values decreased. The study concludes that price negotiations in China effectively reduced drug prices while maintaining a correlation with clinical value, offering insights for price regulation in countries with limited resources and rising drug expenditures. Overall, this is a well-designed and conducted study. While it centers on China, its findings have potential applicability and value for a broader international audience.

Upon thorough examination, it's clear that the authors have effectively incorporated the valuable feedback from the four reviewers. The revisions have enhanced the paper substantially, and I find that it now aligns with the publication's standards. With no additional comments beyond what the reviewers have provided, I recommend accepting the revised manuscript in its current state for publication.

[LINK]

---

## [Editor Report · Decision Letter 2]

8 Dec 2023

Dear Dr. Pan,

Thank you very much for re-submitting your manuscript "Price negotiation and pricing of anticancer drugs in China: An observational study" (PMEDICINE-D-23-02974R2) for review by PLOS Medicine.

I have discussed the paper with my colleagues and the academic editor. I am pleased to say that provided the remaining editorial and production issues are dealt with we are planning to accept the paper for publication in the journal.

[LINK]

We look forward to receiving the revised manuscript by Dec 15 2023 11:59PM.   

Best wishes,

Pippa

Philippa Dodd, MBBS MRCP PhD

PLOS Medicine

plosmedicine.org

pdodd@plos.org

Requests from Editors:

Thank you for your detailed and considered response to previous editor and reviewer requests. Please see below for further comments which we require you address prior to publication.

ABSTRACT

Please provide details in paragraph 1 of the methods and findings section of how you identified the anti-cancer drugs that were negotiated and the trials that reported outcomes – i.e. databases, sites and/or search terms as applicable. Were there any specific exclusions beyond unnegotiated drugs? If so, please also specify these.

AUTHOR SUMMARY

Line 67 – please provide (brief) details of how/where the data were obtained.

Line 79 – perhaps replace ‘and’ with ‘but’ or ‘however’.

Line 83 – suggest ‘These data suggest that implementation…’

INTRODUCTION

We note your response to our previous request regarding SRMA. We know from previous correspondence with you that none exist as per the reported details of the search you conducted in response to the initial peer review of your paper. It would be worth including (brief) details here, which will help to justify your use of individual as opposed to multiple trials in the study.

TABLES

In the footnote of your tables please define ‘CI’.

FIGURES

Thank you for revising your figures and improving their clarity. To me, in the captions of the revised figures 3 and 4 it is not quite as clear as compared to the main text that ‘excluding the extreme outliers and log-transforming treatment costs based on data distribution’ were the adjustments made to the data (at least as I understand things). Many apologies if I have misunderstood but if so, perhaps all the more reason to clarify/revise!

REFERENCES

For in-text reference callouts please remove spaces between different citations. For example line 94, ‘…[1, 2].’ should read, ‘…[1,2].’ Please check and amend throughout.

In the bibliography, please replace ‘cited’ with ‘Accessed’ for web references.

SOCIAL MEDIA

To help us extend the reach of your research, please detail any X (formerly Twitter) handles you wish to be included when we tweet this paper (including your own, your coauthors’, your institution, funder, or lab) in the manuscript submission form when you re-submit the manuscript.

[LINK]

---

## [Editor Report · Decision Letter 3]

13 Dec 2023

Dear Dr Pan, 

On behalf of my colleagues and the Academic Editor, Professor Aaron Kesselheim, I am pleased to inform you that we have agreed to publish your manuscript "Price negotiation and pricing of anticancer drugs in China: An observational study" (PMEDICINE-D-23-02974R3) in PLOS Medicine.

Prior to publication, in the abstract and main manuscript methods and results section, please:

1) define CI and IQR at first use for the reader

2) specify the currency that costs were measured in (US dollars?)

PRESS

Thank you again for submitting to PLOS Medicine, it has been a pleasure handling your manuscript. We look forward to publishing your paper. 

Best wishes,

Pippa 

Philippa Dodd, MBBS MRCP PhD 

PLOS Medicine